# Protective Efficacy of Inactivated H9N2 Vaccine in Turkey Poults under Both Experimental and Field Conditions

**DOI:** 10.3390/vaccines10122178

**Published:** 2022-12-19

**Authors:** Wael K. Elfeil, Hefny Youssef, Ahmed Sedeek, Ahmed El-Shemy, Ehab M. Abd-Allah, Magdy F. Elkady, Eman K. El_Sayed, Abdel-Hamid I. Bazid, Mona S. Abdallah

**Affiliations:** 1Avian and Rabbit Medicine Department, Faculty of Veterinary Medicine, Suez Canal University, Ismailia 41522, Egypt; 2Animal Health Research Institute, Agriculture Research Center, Dokki, Giza 12618, Egypt; 3Parasitology and Animal Disease Department, Veterinary Research Institute, National Research Centre, Dokki, Giza 12622, Egypt; 4Veterinary Hospital, Faculty of Veterinary Medicine, Zagazig University, Zagazig 44511, Egypt; 5Poultry Diseases Department, Faculty of Veterinary Medicine, Beni-Suef University, Beni-Suef 62511, Egypt; 6Virology Department, Faculty of Veterinary Medicine, Suez Canal University, Ismailia 41522, Egypt; 7Virology Department, Faculty of Veterinary Medicine, University of Sadat City, Sadat City 32958, Egypt

**Keywords:** turkey, LPAI H9N2, vaccine efficacy, vaccination, farm vaccination, pathogenicity

## Abstract

Low pathogenic avian influenza (LPAI) H9N2 virus is one of the major poultry pathogens associated with severe economic losses in the poultry industry (broiler, layers, breeders, and grandparents’ flocks), especially in endemic regions including the Middle East, North Africa, and Asian countries. This work is an attempt to evaluate the efficacy of whole inactivated H9N2 vaccine (MEFLUVAC^TM^ H9) in turkey poults kept under laboratory and commercial farm conditions. Here, 10,000 white turkey poults (1-day old) free from maternally derived immunity against H9N2 virus were divided into four groups; G1 involved 10 vaccinated birds kept under biosafety level-3 (BLS-3) as a laboratory vaccinated and challenged group, while G2 had 9970 vaccinated turkeys raised on a commercial farm. Ten of those birds were moved to BLS-3 for daily cloacal and tracheal swabbing to check for the absence of any life-threating disease, before conducting analyses. G3 (10 birds) served as a non-vaccinated challenged control under BSL-3 conditions, while G4 (10 birds) was used as a non-vaccinated and non-challenged control under BSL-3 conditions. Sera were collected on days 7-, 14-, 21-, and 28-post-vaccinations to monitor the humoral immune response using a hemagglutination-inhibition (HI) test. At these same intervals, cloacal and tracheal swabs were also checked for any viral infection. The challenge was conducted 28 days post-vaccination (PV) using AI-H9N2 in BSL-3 by intranasal inoculation of 6-log10 embryo infective dose_50_ (EID_50_). At 3-, 6-, and 10-days post-challenge, oropharyngeal swabs were taken from challenged birds to quantify viral shedding by quantitative polymerase chain reaction (qRT-PCR). The results of this study showed that vaccinated groups (G1/2) developed HI titers of 1.38, 4.38, 5.88, and 7.25 log_2_ in G1 vs. 1.2, 3.8, 4.9 and 6.2 log_2_ in G2 when measured at 7-, 14-, 21- and 28-days PV, respectively, while undetectable levels were recorded in non-vaccinated groups (G3/4). Birds in G3 showed 90% clinical sickness vs. 10% and 20% in G1/2, respectively, over a 10-day monitoring period following challenge. Vaccinated birds showed a significant reduction in virus shedding in terms of the number of shedders, amount of shed virus and shedding interval over the non-vaccinated challenged birds. Regarding mortality, all groups did not show any mortality, which confirms that the circulating H9N2 virus still has low pathogenicity and cannot cause mortality. However, the virus may cause up to 90% clinical sickness in non-vaccinated birds vs. 10% and 20% in laboratory- and farm-vaccinated birds, respectively, highlighting the role of the vaccine in limiting clinical sickness cases. In conclusion, under the current trial circumstances, MEFLUVAC^TM^-H9 provided protective seroconversion titers, significant clinical sickness protection and significant reduction in virus shedding either in laboratory- or farm-vaccinated groups after a single vaccine dose.

## 1. Introduction

The Avian influenza (AI), or Bird flu, is a highly contagious viral disease caused by viruses that are members of the family Orthomyxoviridae. They contain segmented, single-stranded negative sense RNA and may be grouped antigenically into three distinct serotypes, A, B, and C based on internal proteins, principally NP and M1 proteins [1,2,3,4]. Avian influenza viruses (AIVs) fall under influenza virus type-A (IVA) and may be categorized into subtypes depending on antigenic relationships among surface glycoproteins into 18 hemagglutinins (HA) and 11 neuraminidases (NA), with varying permutations [2,5]. According to virulence, AIVs are subdivided into highly pathogenic avian influenza viruses (HPAIV) and low pathogenic avian influenza viruses (LPAIV) [4,6]. LPAIVs have a monobasic hemagglutinin (HA) cleavage site (CS), triggered by trypsin-like enzymes that are primarily present in the respiratory and digestive systems [7]. Because of this, the spread of LPAIV is often limited and the morbidity and mortality rates are lower than those associated with HPAIV, which may induce systemic infections, with failure of numerous organs and up to one hundred percent mortality [8].

In 1966, the first isolation of the AIV H9N2 virus was documented from turkey species in Wisconsin, USA [9]. H9N2 viruses are considered to be the most common AIV subtype in poultry around the world. H9N2 viruses have been found in wild and domestic birds in addition to other species such as horses, ferrets, minks, pigs, and humans during the last 20 years [10,11,12,13,14,15,16,17]. The H9N2 virus has been considered to be one of the major threats to the poultry industry in the Middle East and far Asia countries since the 1990s and several attempts to control the disease have been applied, including preparation of autogenous vaccines, homologues vaccines and then commercial registered vaccines [18]. Since its first official reporting in 2011 and up until now, the H9N2 avian influenza virus has been regarded as one of the primary viral infections plaguing Egypt’s poultry sector [19]. Infection causes severe economic losses due to immunosuppressive effects, complications with other bacterial or viral pathogens, and interfering with other live vaccines [20]. Turkey species are vulnerable to a broad range of type A influenza viruses, including those affecting wild birds, pigs, and humans. As a result, turkeys may serve as a vessel for mixing multiple influenza viruses [21,22,23]. They are susceptible to LPAI H9N2, and some reports indicate that the virus could lead to 5–10% mortality as a single pathogen [23,24,25,26]. Currently, the LPAIV-H9N2 control strategy in the endemic countries in the Middle East and far Asia is based on vaccination of healthy birds with whole inactivated vaccines from different virus groups, vast majority of them belonging to the G- lineage [18].

This work was conducted to evaluate the protective efficacy of a commercial inactivated oil-emulsion H9 vaccine (MEFLUVAC^TM^-H9, produced by the Middle East for Veterinary Vaccines; MEVAC) administered to turkeys raised under both field and experimental laboratory conditions and challenged with a homologous low pathogenic avian influenza virus (H9N2). Variation between seroconversion and protection of birds under those conditions and pathogenicity of recently isolated H9N2 virus in white turkey poults were evaluated.

## 2. Materials and Methods

### 2.1. Birds

The study involved 10,000 white turkey poults (1-day old), which were imported from Aviagen France via a commercial corporate; the birds were treated based on the Egyptian GOVS regulations; birds were free from H9N2 antibodies since their breeder’s flocks did not receive any H9N2 vaccine nor were exposed to infection. The birds were divided into four groups: G1 involved 10 vaccinated birds that were kept under BSL-3 conditions and challenged 28-days post vaccination; G2 had 9970 vaccinated turkeys raised on a commercial farm and at 39 days of age, 10 birds were moved to isolators three days before being challenged. They were monitored daily by tracheal and cloacal swabs using RT-PCR to check for any life threating pathogens through specific primers and probes (unpublished data), including avian influenza (matrix gene) and Newcastle virus. G3 (10 birds): served as a non-vaccinated challenged control and was kept under BSL-3 conditions, while G4 (10 birds) was regarded as a non-vaccinated and non-challenged control (Table 1).

### 2.2. Vaccines

All vaccinated turkeys received the MEFLUVAC^TM^-H9 vaccine, which is a commercially available vaccine prepared from recently circulating virus in the Middle East region and prepared as a water-in-oil emulsion inactivated whole H9N2 vaccine (MEVAC for vaccines, Egypt) at 14 days old as per the manufacturer’s recommendations.

### 2.3. Hemagglutination Inhibition Test

Serum was collected from vaccinated birds at 7-, 14-, 21-, and 28-days post-vaccination (DPV), and antibody levels were measured by the hemagglutination-inhibition (HI) test. as 4 hemagglutinin unit (4HAU), in a V-shape 96-well microplate as per the OIE manual [27].

### 2.4. Challenge Test

Vaccinated birds placed under BSL-3 conditions were challenged at 28 days PV by intranasal inoculation of 6-log_10_ embryo infective dose_50_ (EID_50_) of AI-H9N2 virus in 100 µL phosphate-buffer saline (PBS) on day 28 post vaccination as previously described [24].

### 2.5. qRT-PCR for Virus Detection

Random tracheal and cloacal swabs collected from 10 birds in the G2 (farm group) on 7-, 14-, 21- and 28-days post-vaccination were checked using qRT-PCR for detection of avian influenza (matrix gene), and Newcastle virus (Velogenic strains), using specific primers and probes (unpublished data). Tracheal swabs were collected from the challenged birds for detecting and quantifying virus shedding using RT-PCR with specific primers and probes for LPAIV-H9N2 based on H9 segments at 3-, 6-, 10-, and 12-days post challenge (DPC), as per the OIE manual [27]. The swab samples were subjected to qRT-PCR for virus titration; where standard curves were generated with control viral RNAs and the Ct values of the samples were converted into EID_50_/mL by interpolation as previously described [28,29] A Light Cycler^®^ 96 Real-Time PCR system was used to conduct the qRT-PCR assay (Roche Molecular Bio-chemicals, Mannheim, Germany) using previously validated primers and probes (unpublished data).

## 3. Results

### 3.1. Seroconversion as HI Assay GMT

HI geometric mean titers (GMT) of G1 were 1.38, 4.88, 5. 8, and 7.25 log_2_ when measured at 1-, 2-, 3- and 4-weeks PV, respectively. Titers of G2 were 1.2, 3.5, 4.9, and 6.2 log_2_ at 1-, 2-, 3-, and 4-weeks PV, respectively. Groups 3 and 4 (non-vaccinated groups) did not show detectable antibodies during the same period (Figure 1).

### 3.2. Protection Following Challenge

In relation to clinical sickness, non-vaccinated challenged birds (G-3) showed clinical manifestations, like sneezing, rales, and ruffled feather in around 90% of the birds. Laboratory-vaccinated and challenged birds (G-1) showed clinical signs in only 10% vs. 20% in farm-vaccinated challenged birds (G-2). Regarding mortality, none of the groups showed any mortality either in laboratory-vaccinated, farm-vaccinated, or non-vaccinated challenged birds (Table 2).

### 3.3. Pathogenicity of H9N2 in White Turkey

The non-vaccinated challenged turkeys (white turkeys without any maternally derived immunity) developed clinical manifestations in almost all birds, but without any mortality recorded throughout the 10-day monitoring period post-challenge, while vaccinated birds showed significantly higher sickness protection in the laboratory and farm groups, i.e., 90% and 80%, respectively, (Table 2).

### 3.4. Virus Shedding Following Challenge

The birds in vaccinated groups showed a significant reduction in virus shedding (number of shedders/amounts of virus shedding) in comparison with non-vaccinated challenged birds. Laboratory-vaccinated challenged birds (G-1) showed a significant reduction in virus shedding via the tracheal route 3-DPC and 7-DPC in comparison with farm-vaccinated challenged birds (G-2). However, at 10-DPC on the tracheal route and 3-, 6- and 10-DCP on the cloacal route, the difference between the amount of virus shedding in laboratory-vaccinated (G-1) and farm-vaccinated (G-2) challenged group was not considerable. Birds in G-4 (non-vaccinated non-challenged group) showed non-detectable (ND) virus shedding over the entire monitoring period (Table 3).

### 3.5. Challenge Virus Molecualr Analysis

The challenge virus represented 2021 Egyptian H9N2 isolates, with amino acid matrix identity around 98% in the vaccine seed used for bird vaccination, and the challenge virus hemagglutination-segment amino acids showed a matrix similarity with the common circulating H9N2 in the Middle East, Africa, and south Asia with a range 92–99% (Table 4).

## 4. Discussion

Avian influenza is one of the major threats to the poultry industry worldwide. LPAI viruses are ubiquitous in turkey and chicken species globally, especially in Africa and Asia. Previously it has been reported that turkeys may be more vulnerable to LPAI virus disease than chickens or ducks [30]. Because of the potential losses caused by H9N2 viruses, several countries, such as China, South Korea, Pakistan, Iran, UAE, Morocco, and Egypt, have embraced vaccination as a key strategy for deterring H9N2 disease in poultry [31,32,33,34,35,36]. Conventional inactivated vaccines are most frequently used for protection from clinical disease pictures and reduction of viral shedding [37]. This study revealed that the vaccinated groups under laboratory and farm conditions could induce antibody responses that reached 7.25 log_2_ and 6.2 log_2_, respectively, four weeks post-vaccination Meanwhile, non-vaccinated groups did not show any detectable antibody levels. The observed quicker and higher immune response in turkeys under laboratory conditions compared to farm-raised birds may be associated with differences in management conditions, including ventilation, water quality, space per bird, and feeding space per bird. This is consistent with the previous report of Talat et al., 2020, who recorded that applying single-dose vaccinations with inactivated H9N2 vaccine under laboratory conditions can provide 1–2 log_2_ titers higher than the same vaccine dose under farm conditions [18]. The non-vaccinated and vaccinated infected birds showed no mortality for 10 days post-infection, which confirms that the current circulating AIV-H9N2 virus is of low pathogenicity as previously reported by Elfeil et al., 2018 [24]. Non-vaccinated infected birds showed 90% clinical sickness, indicating that the virus is associated with a pathological picture in white turkey poults that explains the losses in commercial farms due to H9N2 virus itself and upon secondary infection can lead to mortality specifically. Escherichia coli (E. coli) infection has been previously reported by Mahmoud et al., 2022, to cause a severe concurrent infection with H9N2 [37]. Vaccinated birds showed considerable protection against the develop of clinical sickness, portrayed as 10% and 20% in laboratory- and farm-vaccinated groups vs. 90% clinical sickness in non-vaccinated infected birds. This confirms the value of the H9N2 inactivated vaccine in protecting white turkey poults and its value in reducing the clinical picture and losses associated with secondary complications, in agreement with the previous reports of Mahmoud et al. 2022 [37]. However, these findings disagree with another report by Fellahi et al., 2021, who worked on evaluating the pathogenicity of the Moroccan H9N2 in turkeys and the potency of an inactivated commercial vaccine. They found that the mortality rate in unvaccinated turkeys reached up to 50% but decreased to 20% in vaccinated birds. It has been shown that clinical sickness can be exaggerated by secondary infections, specially from commensal bacterial, such as E. coli which may turn clinical sickness into mortality [23,37]. The birds in vaccinated groups, whether under laboratory or farm conditions showed a high reduction in the number of shedders, time of virus shedding, and amount of virus shed, compared to non-vaccinated challenged birds. Similarly, birds kept under laboratory conditions showed a substantial reduction in the number of shedding and amount of virus shedding, but the shedding time in birds kept under farm conditions was not so different, in agreement with the previous report of Elfeil et al., 2019; thus, it may be associated with the close relationship between the challenge virus and the used vaccine as it shares around 98% similarity in the level of hemagglutinin-segment amino-acid structure. These authors observed major differences between laboratory and farm conditions in white turkey following infection with AIV-H9N2 virus, amounting to 7–15% [38]. The close similarity between the vaccine seed and challenge virus with the virus circulating in the selected Middle East, North Africa and Asian countries showed a similarity matrix ranging from 92 to 99%, which may suggest closely related results with those viruses, but to confirm this, there is a need to conduct independent challenge trials in each country with its own isolates.

The results of the current study demonstrated that AIV-H9N2 is still a low-pathogenicity virus that cannot lead to direct mortality as a single pathogen under laboratory conditions; however, it can lead to clinical sickness in white turkeys in around 90% of the birds, which confirmed the higher affinity of H9N2 in white turkeys compared to white broiler chickens as previously report by Elfeil et al. 2018, who demonstrated that the H9N2 virus could lead to clinical sickness in around 30% of infected white broilers [24].

## 5. Conclusions

Based on the obtained results (under laboratory and commercial circumstances), MEFLUVAC^TM^-H9 vaccine provides clinical sickness protection and significant reduction in virus shedding in relation to the number of shedders, shedding virus and time of shedding in white turkeys after a single vaccine dose. A booster dose may be required to maintain the same protection level during a rearing period of 90–120-days. The current circulating H9N2 virus still has low pathogenicity with higher affinity for white turkeys.

## Figures and Tables

**Figure 1 vaccines-10-02178-f001:**
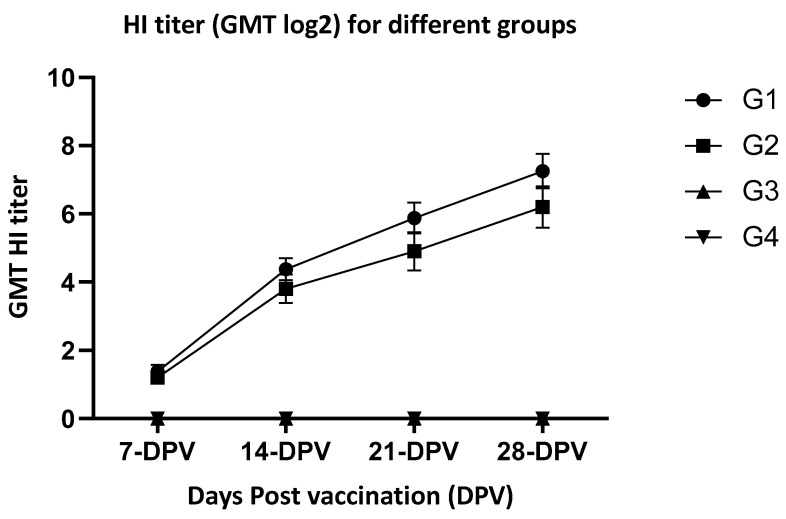
HI titer (GMT log2) for different groups at week 1-, 2-, 3-, 4- post-vaccination. G-1—vaccinated birds under laboratory conditions; G-2—Vaccinated birds under farm conditions; G-3—non-vaccinated birds under laboratory conditions (controls); G-4—non-vaccinated birds under laboratory conditions (controls); and non-detectable level, DPV—days post-vaccination.

**Table 1 vaccines-10-02178-t001:** Experimental design.

Group	Bird No.	Vaccine	Housing	Challenge	Assessment Measure
G-1	10	MEFLUVAC^TM^ H9 at 14th	BSL-3	H9N2	Clinical signs, PM gross lesions, survivability, Virus shedding, Survival rate, qRT-PCR
G-2	9970/10	MEFLUVAC^TM^ H9 at 14th	Farm/BSL-3	H9N2	
G-3	10	Non vaccinated	BSL-3	H9N2	
G-4	10	Non vaccinated	BSL-3	Saline	

G1—vaccinated under BSL-3 and challenged group; G2—birds vaccinated and kept under farm condition and moved to BLS-3 for challenge; G-3—birds kept under BLS-3 conditions as virus control positives, non-vaccinated, challenged; G4—birds kept under BLS-3 conditions used as Control negatives, non-vaccinated, non-challenged.

**Table 2 vaccines-10-02178-t002:** Development of clinical manifestations during 10-days post-challenge.

Group	Clinical Signs	Days Post Challenge										Clinical Protection%	Survivability %
		1	2	3	4	5	6	7	8	9	10		
G-1	Normal	10	10	10	9	9	9	10	10	10	10	9/10 (90%)	10/10 (100%)
	Sick	0	0	0	1	1	1	0	0	0	0		
	Dead	0	0	0	0	0	0	0	0	0	0		
G-2	Normal	10	10	10	9	9	8	10	10	10	10	8/10 (80%)	10/10 (100%)
	Sick	0	0	0	1	1	2	0	0	0	0		
	Dead	0	0	0	0	0	0	0	0	0	0		
G-3	Normal	10	10	9	6	3	3	6	8	9	10	1/10 (10%)	10/10 (100%)
	Sick	0	0	1	4	7	7	4	2	1	0		
	Dead	0	0	0	0	0	0	0	0	0	0		
G-4	Normal	10	10	10	10	10	10	10	10	10	10	10/10 (100%)	10/10 (100%)
	Sick	0	0	0	0	0	0	0	0	0	0		
	Dead	0	0	0	0	0	0	0	0	0	0		

G-1—vaccinated birds under laboratory conditions; G-2—vaccinated birds under farm conditions; G-3—non-vaccinated birds under laboratory conditions (controls); G-4—non-vaccinated birds under laboratory conditions (controls); and non-detectable level, DPV—days post-vaccination.

**Table 3 vaccines-10-02178-t003:** Virus shedding for different groups at week 3, 6, 10 days post-infection.

Gr. No.	Virus Shedding on 3, 6, 10 DPC via Cloacal and Tracheal Route Assessment					
	3-DPC		6-DPC		10-DPC	
	Tracheal Swabs	Cloacal Swabs	Tracheal Swabs	Cloacal Swabs	Tracheal Swabs	Cloacal Swabs
	No./EID_50_	No./EID_50_	No./EID_50_	No./EID_50_	No./EID_50_	No./EID_50_
G-1	3/10 (2.1 ± 0.4) ^a^	2/10 (1.9 ± 0.4) ^a^	1/10 (1.4 ± 0.0) ^a^	2/10 (2.3 ± 0.4) ^a^	nd	nd
G-2	4/10 (2.6 ± 0.5) ^b^	3/10 (2.1 ± 0.5) ^a^	2/10 (1.9 ± 0.3) ^b^	3/10 (2.9 ± 0.5) ^a^	nd	nd
G-3	10/10 (4.2 ± 0.9) ^c^	6/10 (2.9 ± 0.5) ^b^	7/10 (3.8 ± 0.9) ^c^	10/10 (5.2 ± 0.8) ^b^	3/10 (2.2 ± 0.4) ^b^	5/10 (3.6 ± 0.6) ^b^
G-4	nd	nd	nd	nd	nd	nd

Gr. No.—Group Number; G-1—laboratory-vaccinated challenged birds; G-2—farm-vaccinated challenged birds; G-3—laboratory non-vaccinated challenged birds; G-4—laboratory non-vaccinated non-challenged birds; DPC—days post challenge; EID_50_—Egg infected dose_50_; nd—not detected under the used PCR test condition and threshold. ^a, b, c^ describes the significance difference in-between the groups.

**Table 4 vaccines-10-02178-t004:** Challenge virus hemagglutination gene amino acid similarity matrix.

	Virus	Vac	EG11	SA98	IR98	AE99	EG21	AE15	QR08	JO20	LB18	SA18	IQ16	LY06	DZ17	MR20	TN18	UG19	TG19	NG19	KY21	BJ20	BD22	NP11	PK21	IR18	IN20	RU18
**Virus**		98.2	97.1	93.3	93.1	92.5	99.4	94.3	95.8	93.9	95.1	93.3	92.8	95.8	94.3	94.5	95.2	94.1	92.9	93.3	93.7	93.3	93.7	93.3	92.9	93.7	92.4	96
**Vac**	98.2		98.4	93.2	93	93.1	97.6	95.2	96.6	94.8	95.8	94.2	92.5	96.6	95.2	95.4	96	95.8	93.8	93.8	95.4	93.8	93.6	93.4	93.2	93.4	92.6	96.8
**EG11**	97.1	98.4		94.2	93.9	93.3	96.4	95.1	96.4	94.4	95.2	94.6	92.8	96.8	95.1	94.8	96.2	95.1	93.7	93.9	94.8	93.9	93.9	93.7	93.1	93	92.6	97.6
**SA98**	93.3	93.2	94.2		98.6	97.8	92	92.5	94.1	91.8	91.2	92.2	91.3	94.8	92.5	92	94.3	92	91.3	91.8	91.6	91.6	91.8	93	91.4	91.8	90.4	94.8
**IR98**	93.1	93	93.9	98.6		97.1	91.4	92	93.6	91.3	91.2	91.9	90.7	93.9	92.1	91.6	93.6	91.8	91.1	91.4	91.4	91.3	91.1	92.3	90.5	91.6	89.8	94.4
**AE99**	92.5	93.1	93.3	97.8	97.1		90.8	92.1	93.6	91.2	91.8	91.5	90.1	94.3	92.1	91.4	93.8	91.4	91	91	91	90.8	91.2	92.3	90.1	91.2	88.8	94.5
**EG21**	99.4	97.6	96.4	92	91.4	90.8		93.2	94.6	92.5	93.2	92.6	91.4	94.6	93.2	93.4	94.1	93	92	92.3	92.7	92.3	92.5	92.1	91.4	91.6	90	94.8
**AE15**	94.3	95.2	95.1	92.5	92	92.1	93.2		97.7	96.8	94.2	96.4	93	96.1	99.5	98.4	95.4	96.8	97.9	98.2	96.4	98	92.5	93.2	93.2	93.2	90.4	95.6
**QR08**	95.8	96.6	96.4	94.1	93.6	93.6	94.6	97.7		96.6	96.6	96.6	93.8	98	97.7	97	97.3	97.3	96.3	96.4	97	96.3	93.9	94.5	94.1	93.9	91.6	97
**JO20**	93.9	94.8	94.4	91.8	91.3	91.2	92.5	96.8	96.6		93.5	97.3	91.8	95.4	96.6	96.3	95.5	95.5	95.4	95.5	95.2	95.4	92.5	92.1	92.5	92	90.2	95.2
**LB18**	95.1	95.8	95.2	91.2	91.2	91.8	93.2	94.2	96.6	93.5		92.9	90.1	96.6	93.9	92.9	94.9	93.9	91.8	92.9	93.5	92.5	93.5	92.5	91.5	92.5	90.5	95.9
**SA18**	93.3	94.2	94.6	92.2	91.9	91.5	92.6	96.4	96.6	97.3	92.9		91.3	95.5	96.2	95.9	95.7	95.3	95	95.1	95	95	92.4	92.2	91.7	91.7	90.8	95.5
**IQ16**	92.8	92.5	92.8	91.3	90.7	90.1	91.4	93	93.8	91.8	90.1	91.3		93.9	92.9	92.7	93.2	92.3	92	92.3	92.1	92.3	91.3	92	95.9	96.4	89.8	92.8
**LY06**	95.8	96.6	96.8	94.8	93.9	94.3	94.6	96.1	98	95.4	96.6	95.5	93.9		96.1	95.4	98.6	96.1	94.6	94.8	95.7	94.6	94.5	95	94.3	93.9	92.7	97.4
**DZ17**	94.3	95.2	95.1	92.5	92.1	92.1	93.2	99.5	97.7	96.6	93.9	96.2	92.9	96.1		98.4	95.4	96.6	97.7	98	96.3	97.9	92.5	93.2	93	93	90.4	95.6
**MR20**	94.5	95.4	94.8	92	91.6	91.4	93.4	98.4	97	96.3	92.9	95.9	92.7	95.4	98.4		94.6	96.4	96.8	97.1	96.1	97	92	92.9	92.8	92.7	89.6	94.8
**TN18**	95.2	96	96.2	94.3	93.6	93.8	94.1	95.4	97.3	95.5	94.9	95.7	93.2	98.6	95.4	94.6		95.5	94.3	94.1	95.2	93.9	93.9	94.6	93.2	93.2	92.5	96.9
**UG19**	94.1	95.8	95.1	92	91.8	91.4	93	96.8	97.3	95.5	93.9	95.3	92.3	96.1	96.6	96.4	95.5		95.2	95.5	99.6	95.4	92.3	92.7	92.8	92.5	90.4	95
**TG19**	92.9	93.8	93.7	91.3	91.1	91	92	97.9	96.3	95.4	91.8	95	92	94.6	97.7	96.8	94.3	95.2		97.7	94.8	97.5	91.1	92.1	92.1	92.1	89.5	94.1
**NG19**	93.3	93.8	93.9	91.8	91.4	91	92.3	98.2	96.4	95.5	92.9	95.1	92.3	94.8	98	97.1	94.1	95.5	97.7		95.2	99.8	92	93	92.5	92.5	89.8	94.3
**KY21**	93.7	95.4	94.8	91.6	91.4	91	92.7	96.4	97	95.2	93.5	95	92.1	95.7	96.3	96.1	95.2	99.6	94.8	95.2		95	92	92.3	92.7	92.3	90.2	94.6
**BJ20**	93.3	93.8	93.9	91.6	91.3	90.8	92.3	98	96.3	95.4	92.5	95	92.3	94.6	97.9	97	93.9	95.4	97.5	99.8	95		92	92.9	92.3	92.3	89.8	94.1
**BD22**	93.7	93.6	93.9	91.8	91.1	91.2	92.5	92.5	93.9	92.5	93.5	92.4	91.3	94.5	92.5	92	93.9	92.3	91.1	92	92	92		92.9	92	91.4	91.3	93.9
**NP11**	93.3	93.4	93.7	93	92.3	92.3	92.1	93.2	94.5	92.1	92.5	92.2	92	95	93.2	92.9	94.6	92.7	92.1	93	92.3	92.9	92.9		92.1	92.3	91.1	94.1
**PK21**	92.9	93.2	93.1	91.4	90.5	90.1	91.4	93.2	94.1	92.5	91.5	91.7	95.9	94.3	93	92.8	93.2	92.8	92.1	92.5	92.7	92.3	92	92.1		96.2	90	93.3
**IR18**	93.7	93.4	93	91.8	91.6	91.2	91.6	93.2	93.9	92	92.5	91.7	96.4	93.9	93	92.7	93.2	92.5	92.1	92.5	92.3	92.3	91.4	92.3	96.2		88.8	93.7
**IN20**	92.4	92.6	92.6	90.4	89.8	88.8	90	90.4	91.6	90.2	90.5	90.8	89.8	92.7	90.4	89.6	92.5	90.4	89.5	89.8	90.2	89.8	91.3	91.1	90	88.8		91.9
**RU18**	96	96.8	97.6	94.8	94.4	94.5	94.8	95.6	97	95.2	95.9	95.5	92.8	97.4	95.6	94.8	96.9	95	94.1	94.3	94.6	94.1	93.9	94.1	93.3	93.7	91.9	

Virus—challenge virus; vac—used vaccine”MEFLUVAC-H9”; IR98—Iran H9N2 1998 isolate; SA98—Kingdom of Saudi Arabia H9N2 1998 isolate; AE99—United Arab Emirates H9N2 1999 isolate; EG21—Egyptian H9N2 2021 isolate; AE15—United Arab Emirates H9N2 2015 virus; QA08—Qatar H9N2 2008 virus; JO20—Jordan H9N2 2020 isolate; LB18—Lebanon 2018 H9N2 isolate; SA16—Kingdom of Saudi Arabia H9N2 2016 isolate; KW04—Kuwait H9N2 2004 isolate; IQ16—Iraq H9N2 2016 isolate; LY06—Libya H9N2 2006 isolate; DZ17—Algeria H9N2 2017 isolate; MA20—Morocco H9N2 2020 isolate; TN18—Tunisia H9N2 2018 isolate; UG19—Uganda H9N2 2019 isolate; TG19—Togo H9N2 2019 isolate; NG19—Nigeria H9N2 2019 isolate; KY21—Kenya H9N2 2021 isolate; BJ22Benin H9N2 2022 isolate; BD22—Bangladesh H9N2 2022 isolate; NP11—Nepal H9N2 2011 isolate; PK21—Pakistan H9N2 2021 isolate; IR18—Iran H9N2 2018 isolate; IN20—India H9N2 2020 isolate; RU18—Russia H9N2 2018 isolate.

## Data Availability

All data are included in the manuscript.

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
