# Peer review of "Protective Efficacy of Inactivated H9N2 Vaccine in Turkey Poults under Both Experimental and Field Conditions"

_vaccines, 2022, doi:10.3390/vaccines10122178_

Round 1

Reviewer 1 Report

The authors conducted several experiments to evaluate the protective efficacy of inactivated H9N2 vaccine in turkey poults, and the results are satisfactory. However, the manuscript needs to be carefully revised to improve its clarity and the data presentation.

1. Line 25, the ‘.’ after ‘3 days’ should be deleted.

2. There are too many contents in Abstract, the author should better to delete redundant sentences.

3. Line 63, ‘In 1966….’ should be started with a new paragraph.

4. Table 2, the data of antibody level may be presented with a line chart instead of table, which could be more visualized.

Author Response

Comments and Suggestions for Authors

The authors conducted several experiments to evaluate the protective efficacy of inactivated H9N2 vaccine in turkey poults, and the results are satisfactory. However, the manuscript needs to be carefully revised to improve its clarity and the data presentation.

  1. Line 25, the ‘.’ after ‘3 days’ should be deleted.

Response: deleted

  1. There are too many contents in Abstract, the author should better to delete redundant sentences.

Response: concise

  1. Line 63, ‘In 1966….’ should be started with a new paragraph.

Response: started new paragraph  

  1. Table 2, the data of antibody level may be presented with a line chart instead of table, which could be more visualized.

Response: changed to line chart

Reviewer 2 Report

In the work the authors use a commercial H9 vaccine (MEF-LUVACTM ) to vaccinate and challenge turkey poults with H9N2. The H9N2 virus used in this case was a variant that caused low pathogenicity in the challenged birds. Birds were vaccinated at two weeks of age and challenged four weeks later (at 6 weeks of age).

The authors found that the vaccinated birds had high HI counts post vaccination although it is difficult to compare the value for these HI numbers with vaccinations Interestingly, only 10% and 20% of challenged vaccinated birds showed clinical sickness. Unvaccinated birds did not have antibody HI counts post vaccination and 90% showed clinical sickness after challenge. Vaccinated birds shed less virus than unvaccinated birds. This number should be given in the abstract ie 2-fold lower shedding.

These results are interesting. It would be useful to know if vaccination protected against egg lay losses post infection and if any protection was awarded if the challenge was carried out several weeks post vaccination. 

My major criticism is that the title and legends for Table 4 (line 169 mislabelled as Table 2) are not clear. The title states that shedding was assayed post-vaccination but I think the authors meant the title to be ‘post infection/post challenge.’ Furthermore, the Table legend should also state that nd = not detected. If this assay was ‘not done,’ nd; then this experiment is meaningless. If it was done, then turkey poults that are infected with H2N9 at 6 weeks of age shed much less virus (2-fold difference?) than control birds which is an interesting result. However, were any mutations detected in the virus shed by the vaccinated poults?

Author Response

Comments and Suggestions for Authors

In the work the authors use a commercial H9 vaccine (MEFLUVACTM ) to vaccinate and challenge turkey poults with H9N2. The H9N2 virus used in this case was a variant that caused low pathogenicity in the challenged birds. Birds were vaccinated at two weeks of age and challenged four weeks later (at 6 weeks of age).

The authors found that the vaccinated birds had high HI counts post vaccination although it is difficult to compare the value for these HI numbers with vaccinations Interestingly, only 10% and 20% of challenged vaccinated birds showed clinical sickness. Unvaccinated birds did not have antibody HI counts post vaccination and 90% showed clinical sickness after challenge. Vaccinated birds shed less virus than unvaccinated birds. This number should be given in the abstract ie 2-fold lower shedding.

These results are interesting. It would be useful to know if vaccination protected against egg lay losses post infection and if any protection was awarded if the challenge was carried out several weeks post vaccination. 

Response: evaluating the vaccine in white turkey breeders and evaluate the effect on egg production, sure will be great add, we will consider it in future work

My major criticism is that the title and legends for Table 4 (line 169 mislabelled as Table 2) are not clear. The title states that shedding was assayed post-vaccination but I think the authors meant the title to be ‘post infection/post challenge.’ Furthermore, the Table legend should also state that nd = not detected. If this assay was ‘not done,’ nd; then this experiment is meaningless. If it was done, then turkey poults that are infected with H2N9 at 6 weeks of age shed much less virus (2-fold difference?) than control birds which is an interesting result. However, were any mutations detected in the virus shed by the vaccinated poults?

Response: the title miswrote, it corrected, nd = not detected “added as table note”

There is mutation in the virus, the molecular analysis data addressed elsewhere

Included hemagglutination gene amino acid similarity index between the challenge virus and the used vaccine as table 4

Table 4. challenge virus hemagglutination gene amino acid similarity matrix

Virus: challenge virus; vac: used vaccine”MEFLUVAC-H9”; IR98: Iran H9N2 1998 isolate; SA98: Kingdom of Saudi Arabia H9N2 1998 isolate; AE99: United Arab Emirates H9N2 1999 isolate;  EG21: Egyptian H9N2 2021 isolate; AE15: United Arab emirates H9N2 2015 virus; QA08: Qatar H9N2 2008 virus; JO20: Jordan H9N2 2020 isolate; LB18: Lebanon 2018 H9N2 isolate; SA16: Kingdom of Saudi Arabia H9N2 2016 isolate; KW04: Kuwait H9N2 2004 isolate; IQ16: Iraq H9N2 2016 isolate; LY06: Libya H9N2 2006 isolate; DZ17: Algeria H9N2 2017 isolate; MA20: Morocco H9N2 2020 isolate; TN18: Tunisia H9N2 2018 isolate; UG19: Uganda H9N2 2019 isolate; TG19: Togo H9N2 2019 isolate; NG19: Nigeria H9N2 2019 isolate; KY21: Kenya H9N2 2021 isolate; BJ22: Benin H9N2 2022 isolate; BD22: Bangladesh H9N2 2022 isolate; NP11: Nepal H9N2 2011 isolate; PK21: Pakistan H9N2 2021 isolate; IR18: Iran H9N2 2018 isolate; IN20: India H9N2 2020 isolate; RU18: Russia H9N2 2018 isolate.

Reviewer 3 Report

In the manuscript by Elfeil et.al, the authors have investigated the efficacy of an inactivated H9 vaccine (MEF-LUVACTM H9) in turkey poults under laboratory and commercial farm conditions. The H9N2 avian influenza virus causes significant threats to the poultry industry globally. Also, it represents a potential threat to human health through its high rate of zoonotic infection.

The approach used by an author to investigate vaccine efficacy provided a better understanding of the protective efficacy of investigated vaccines. I have a few concerns as described below: -

Concern:  

Introduction: Authors can improve the introduction section by including the status of H9N2 worldwide, and currently used control and vaccination strategies by various countries.

Materials and Methods:

Even though OIE protocols were followed to perform HI, it would be great if authors can briefly describe the methods, which will give quick glance to the reader about the procedure.

Also, please provide more detail about the vaccine formulation.

In the case of RT-qPCR, include the primer lists

Results:

Table 2 and Table 4 are given the same heading, please change the appropriate title for Table 4.

Important: Extensive spell check is required, also remove the unnecessary template description at the start of the result section.

Author Response

Comments and Suggestions for Authors

In the manuscript by Elfeil et.al, the authors have investigated the efficacy of an inactivated H9 vaccine (MEF-LUVACTM H9) in turkey poults under laboratory and commercial farm conditions. The H9N2 avian influenza virus causes significant threats to the poultry industry globally. Also, it represents a potential threat to human health through its high rate of zoonotic infection.

The approach used by an author to investigate vaccine efficacy provided a better understanding of the protective efficacy of investigated vaccines. I have a few concerns as described below: -

Concern:  

Introduction: Authors can improve the introduction section by including the status of H9N2 worldwide, and currently used control and vaccination strategies by various countries.

Response: added “

Line 67-70: The H9N2 virus consider one of the major threats to poultry industry in the middle east and far Asia countries since the 1990s and several attempts to control the disease ap-plied including preparation autogenous vaccines, homologues vaccines then commercial registered vaccines.

Line 78-81: Currently, the LPAIV-H9N2 control strategy in the endemic countries in the middle east and far Asia region based on vaccination healthy birds with whole inactivated vaccines from different virus groups, vast majority of it belong to G- lineage

Materials and Methods:

Even though OIE protocols were followed to perform HI, it would be great if authors can briefly describe the methods, which will give quick glance to the reader about the procedure.

Also, please provide more detail about the vaccine formulation.

Response: added some public released details “the vaccine is commercially available vaccine prepared as water in oil emulsion inactivated whole H9N2 vaccine”

In the case of RT-qPCR, include the primer lists

Response: the primers is part from another trial to validate diagnostic set for different H9N2 virus groups

Results:

Table 2 and Table 4 are given the same heading, please change the appropriate title for Table 4.

 Response: changed

Important: Extensive spell check is required, also remove the unnecessary template description at the start of the result section.

Response: removed
